



# The signature of the tropospheric gravity wave background in observed mesoscale motion

**Claudia Christine Stephan**[1] **and Alexis Mariaccia**[2,a]

[1]Max Planck Institute for Meteorology, Hamburg, Germany
[2]Département de Physique, Université Claude Bernard Lyon 1, Villeurbanne, France
[a]now at: Laboratoire Atmosphères, Observations Spatiales, Guyancourt, France

**Correspondence:** Claudia Christine Stephan (claudia.stephan@mpimet.mpg.de)

**Abstract.** How convection couples to mesoscale vertical motion and what determines these motions is poorly understood. This study diagnoses profiles of area-averaged mesoscale divergence from measurements of horizontal winds collected by an extensive upper-air sounding network of a recent campaign over the western tropical North Atlantic, the Elucidating the Role of Clouds-Circulation Coupling in Climate (EUREC⁴A) campaign. Observed area-averaged divergence amplitudes scale approximately inversely with area-equivalent radius. This functional dependence is also confirmed in reanalysis data and a global, freely evolving simulation run at 2.5 km horizontal resolution. Based on the numerical data it is demonstrated that the energy spectra of inertia gravity waves can explain the scaling of divergence amplitudes with area. At individual times, however, few waves can dominate the region. Nearly monochromatic tropospheric waves are diagnosed in the soundings by means of an optimized hodograph analysis. For one day, results suggest that an individual wave directly modulated the satellite-observed cloud pattern. However, because such immediate wave impacts are rare, the systematic modulation of vertical motion due to inertia–gravity waves may be more relevant as a convection-modulating factor. The analytic relationship between energy spectra and divergence amplitudes proposed in this article, if confirmed by future studies, could be used to design better external forcing methods for regional models.

## 1 Introduction

The effects of large-spatial-scale and long-timescale processes on convection are well understood, such as shallow convection prevailing in geographic locations of free-tropospheric subsidence associated with the Hadley cell (e.g., Albrecht et al., 1995; Norris, 1998; Wood and Hartmann, 2006). These links between convective regimes and circulation averaged over seasonal timescales are often the basis for prescribing an external forcing in limited-area models (Albrecht et al., 1979; Sobel and Bretherton, 2000; van der Dussen et al., 2016). Cloudiness in conventional climate models is highly sensitive to the seasonal and synoptic-scale variations in the atmospheric state, but clouds do not vary adequately on timescales shorter than a day (e.g., Nuijens et al., 2015). Whereas models struggle to capture sub-daily cloud variability, which is pronounced on scales of a few tens to several hundreds of kilometers, several observational studies have established direct links between sub-daily variability of cloudiness and mesoscale – i.e., $\mathcal{O}(100)$ km – vertical motion. For instance, Vogel et al. (2020) found that the shallow convective mass flux is regulated primarily by mesoscale vertical motion at cloud base. George et al. (2020) showed that lower-tropospheric mesoscale vertical velocity explains a larger fraction of variations in cloud base cloud fraction than local humidity or stability. Mauger and Norris (2010) reported variations in near-surface divergence to be the most important driver of cloud cover in the subtropical eastern North Atlantic on timescales shorter than 12 h, whereas boundary layer stratification was relevant for timescales of 12–48 h. Their study relied on satellite observations as well as on reanalysis data to complement the me-

teorological fields. The still relatively poor understanding of how clouds couple to circulation on scales of 20–200 km was the main motivation for recent efforts to design observations to measure the vertical profiles of area-averaged divergence on sub-daily timescales (e.g., Bony et al., 2017).

By deriving vertical profiles of divergence from sondes dropped along circular flight paths of 200 km diameter, Bony and Stevens (2019) showed that wave-like structures with vertical wavelengths of several kilometers, autocorrelations of multiple hours, and amplitudes on the order of $2 \times 10^{-5}\,\mathrm{s}^{-1}$ dominated the divergence structure in the free troposphere over the tropical Atlantic. Stephan et al. (2020) confirmed such features in soundings from the 2006 Tropical Warm Pool–International Cloud Experiment (TWP-ICE; May et al., 2008), which took place near Darwin, Australia. Furthermore, they tested if the characteristics of divergence variability might be consistent with the gravity wave dispersion relation and concluded that gravity waves may serve to explain the temporal, vertical, and horizontal variability observed in area-averaged horizontal divergence. Their study, however, is limited in two ways. First, they only diagnosed the average properties of gravity waves passing the domain, not how variable the wave characteristics were. Second, the TWP-ICE campaign, as well as the circle flights of Bony and Stevens (2019), used constant-sized areas with a diameter of about 200 km and thus could not be used to quantify the dependence of divergence characteristics on area size.

In this study we analyze the extensive radiosounding network (Stephan et al., 2021) of the Elucidating the Role of Clouds-Circulation Coupling in Climate (EUREC[4]A) campaign (Stevens et al., 2021). A network composed of a station at Barbados, four ships, and two aircraft collected more than 2700 vertical profiles of winds, temperature, and relative humidity over the course of 33 d. Since the ships were moving, this network sampled area-averaged divergence over different-sized regions. We derive area-averaged divergence using a three-dimensional variational analysis (described in Sect. 2.2). Independently of this, we diagnose the properties of gravity waves by applying a hodograph analysis to the horizontal winds measured by radiosondes (described in Sect. 2.3).

One aim of the present study is to interpret the properties of mesoscale divergence variability in the context of the tropospheric background wave spectrum, which we diagnose from the ERA5 reanalysis (Hersbach et al., 2020) and a global 2.5 km horizontal-resolution simulation initialized on the starting date of the campaign. It is well known that the global kinetic energy spectrum follows a $k^{-3}$ power law at synoptic scales of 1000–4000 km (e.g., Boer and Shepherd, 1983), which is related to slow balanced dynamics, and a $k^{-5/3}$ power law at mesoscales (e.g., Nastrom and Gage, 1985), which is mostly associated with fast unbalanced dynamics and divergence. Hence, one may expect a link between the spectral characteristics of the global energy spectrum and the characteristics of regional area-averaged divergence. A related question is to what extent the tropospheric wave background at given times and locations is saturated versus being dominated by nearly monochromatic waves. Several case studies have shown that individual gravity waves can organize convection (e.g., Shige and Satomura, 2001; Stechmann and Majda, 2009; Lane and Zhang, 2011). Therefore, a second aim of this study is to extract the properties of individual gravity waves from the soundings to quantify their variability, and to test if there are any direct wave imprints visible in patterns of convection.

In Sect. 2 we describe the observations, numerical data, and methods of our analyses. The questions of if and how global energy spectra are linked to mesoscale divergence are discussed in Sect. 3.1, whereas Sect. 3.2 focuses on local waves at specific times. Results are summarized and discussed in Sect. 4.

## 2 Data and methods

### 2.1 Observational data

To discuss potential links between vertical profiles of mesoscale divergence and convection, we show images from the Geostationary Operational Environmental Satellite-16 (GOES-16), which is centered on 75.2° W. Infrared (10.3 μm, Band 13) images of the GOES-16 Advanced Baseline Imager (ABI; Schmit et al., 2017) are provided by NASA's Worldview application (https://worldview.earthdata.nasa.gov, last access: 24 November 2020). Information on the cloud bottom and top levels derived from the GOES-16 ABI (NOAA, 2018) is distributed by the Langley SATCORPS group of NASA.

Diagnostics of mesoscale divergence and gravity wave characteristics are based on measurements of surface-launched radiosondes, and dropsondes released from aircraft, taken during the EUREC[4]A campaign in January and February 2020. The experimental design of EUREC[4]A involved 85 dropsonde circles from aircraft flights and regular releases of surface-launched radiosondes from five platforms. A combined data set with 895 dropsonde soundings from the German Aerospace Center's HALO aircraft and 322 dropsonde soundings from NOAA's WP-3D will be made available as part of the Joint dropsonde-Observations of the Atmosphere in tropical North atlaNtic large-scale Environments (JOANNE) data set described elsewhere. We use the quality-controlled Level 2 data with a vertical resolution of 10 m. HALO's circle typically consisted of 12 sondes and is centered at $\sim 13.3°$ N, $-57.7°$ E, with a diameter of 200 km, as shown in Fig. 1b. The WP-3D circles, of which there are only 14, were mostly flown further east at varying locations within 11–16° N, 59–50° W. HALO data cover the surface up to the flight level of 9 km; WP-3D data cover the surface up to lower and variable flight levels with a maximum altitude of 7 km. We ingest all available dropsonde and

## (a) temporal coverage

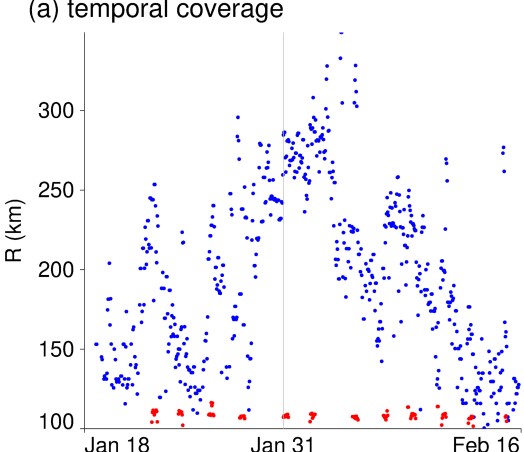

## (b) polygon Jan 31, 16 UTC

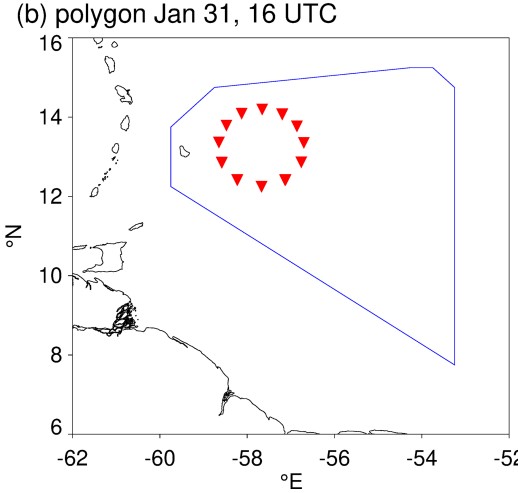

**Figure 1. (a)** Equivalent radius of the areas sampled by the sounding network (blue) and the HALO circles (red) over the period of the campaign. All points fulfill $L/R < 2$ (see the text for details). **(b)** For 31 January 2020 at 16:00 UTC an example of a polygon spanned by radiosoundings (blue) and the location of dropsondes from the HALO circle (red).

surface-launched soundings into a three-dimensional variational analysis (3D-Var; described in Sect. 2.2) to estimate the area-averaged divergence. A separate 3D-Var analysis is performed just on the HALO circles to estimate divergence on this smaller scale.

In addition to using them in 3D-Var, we analyze Level 2 surface-launched soundings for gravity waves. These data, which cover the ground to the stratosphere (typically 25 km) in steps of 10 m, are described in Stephan et al. (2021) and are publicly available at https://doi.org/10.25326/62. Unlike the circle flights, the continuous data coverage allows us to filter them in time to a specific frequency interval, as we describe in Sect. 2.3. These radiosondes were launched regularly (usually 4 hourly) from a network consisting of the Barbados Cloud Observatory (BCO) and four ships within 6–16° N,

60–51° W: the R/V *L'Atalante*, R/V *MS-Merian*, R/V *Meteor*, and R/V *RH-Brown*. From 8 January to 19 February, the network launched a total of 848 radiosondes from the surface and also recorded the descent for 82 % of them. Since a typical ascent took about 90 min, this resulted in high-frequency coverage during normal operations. The *Meteor* remained nearly stationary at a longitude of 57° W and moved north- and southward between 12.0–14.5° N. The *RH-Brown* moved approximately orthogonal to *Meteor*'s sampling line between the BCO (13.16° N, 59.43° W) and the Northwest Tropical Atlantic Station for air–sea flux measurements buoy (NTAS) at 14.82° N, 51.02° W. The *MS-Merian* and *Atalante* ventured further to the south to a minimum latitude of $\sim 6.5°$ N and were often in close proximity, i.e., $< 200$ km apart. The ships' routes are displayed in Fig. 2 of Stephan et al. (2021).

### 2.2 3D variational analysis

The computation of divergence and vorticity from unfiltered sounding data is performed with an analysis tool called 3D-Var, described in López Carrillo and Raymond (2011). It ingests spatially nonuniform profiles of temperature, horizontal wind, mixing ratio, and pressure and interpolates them onto a regular grid of the user's choice. Following a weighted least-squares algorithm, 3D-Var then minimizes the misfit between the computed fields and the gridded input measurements. This minimization problem is additionally constrained by the anelastic mass continuity equation. We use a $0.5° \times 0.5°$ longitude–latitude grid for analyzing data from the full network of soundings and a $0.1° \times 0.1°$ longitude–latitude grid for our separate analysis of data from HALO's circle flights. Since we are interested in variability on vertical scales $> 1$ km, a vertical grid spacing of 200 m is chosen, covering the surface up to 16 km.

3D-Var is applied every hour between 18 January and 16 February (720 h). From the analysis of Bony and Stevens (2019) divergence autocorrelations for the HALO circles are expected to exceed 3 h and to increase with the area over which divergence is averaged. Therefore, to improve data density, we ingest, for each given time, data within $\pm(\Delta t = 90$ min). This choice results in nearly continuous 3D-Var results between 18 January and 16 February, as can be seen in Fig. 1a, which shows the equivalent radius, computed as $R = \sqrt{A/\pi}$. The area $A$, in which the data assimilation is performed, is limited to the region with good data coverage, to avoid extrapolation. For each application of 3D-Var, this area is defined by the boundaries of a polygon that encloses all available soundings and whose corners are grid points with full vertical coverage in all variables. For 31 January, Fig. 1b shows an example of such a polygon. The blue polygon is the mask for the $0.5° \times 0.5°$ all-soundings assimilation, and the red triangles mark the polygon bounds of the $0.1° \times 0.1°$ HALO-circle assimilation. We discard all times at which the aspect ratio of a polygon's perimeter $L$ and the equivalent radius $R$ (as defined above) is $L/R > 2$, to limit

the analysis to areas with little distortion. For the blue polygon in Fig. 1b, for instance, $L/R = 1.3$. The output of 3D-Var contains the 3D fields of divergence and vorticity within the polygon bounds. We average them horizontally to compute vertical profiles of divergence and vorticity.

## 2.3  Hodograph analysis

The hodographs of the horizontal wind perturbations associated with low-frequency gravity waves describe ellipses whose properties can be determined using the Stokes parameters, as in Eckermann and Vincent (1989). Properties that follow immediately are the ratio of the minor to the major axis, $\epsilon$; the orientation of the major axis relative to the zonal direction, $\Theta$; and the degree of polarization of the ellipse, $d$. Large $d$ indicates that a sounding contains coherent waves rather than noise. Therefore, we iteratively search for time–altitude windows in the radiosonde measurements that give large values of $d$, as described in the following.

We analyze data from each platform separately, with one exception: over the studied period, the *MS-Merian* launched fewer radiosondes (118) than the others ships (*Meteor*: 203; *RH-Brown*: 169), which was partly due to her proximity to the *Atalante* (139 radiosondes). Thus, to achieve continuous data coverage, we include selected soundings collected by the *MS-Merian* with those of the *Atalante*; specifically, soundings of the *MS-Merian* that were taken within 100 km of the *Atalante* are added to the *Atalante*'s data set. This choice is reasonable as the drift of a sonde can be on the order of 100 km. From now on, we will refer to this merged data set as the combined data set. The BCO is the only spatially fixed platform. They launched 182 sondes, whose horizontal wind measurements from 21 January at 02:00 UTC to 17 February at 12:00 UTC are shown in Fig. 2a and b.

Missing values in each data set are filled by interpolating in time to a 1-hourly resolution. A Lanczos filter is applied to horizontal winds and temperature profiles to retain vertical wavelengths of $500 \, \text{m} < \lambda_z < 10 \, \text{km}$ and periods of $3 \, \text{h} < \tau < 53 \, \text{h}$, where 53 h corresponds to the inertial period at the BCO. Figure 2c and d show the resulting horizontal wind perturbations for the BCO. Coherent structures are clearly visible in both wind components. Oftentimes they exhibit a downward propagation with time, which is characteristic of upward-propagating gravity waves.

At each time we compute the quadrature spectrum between the zonal and meridional wind perturbations, $u'(z)$ and $v'(z)$. Then we search the resulting spectrum, which is a function of time and vertical wave number $m$, for local maxima in $m$ that persist in time for at least 10 h. The vertical profiles of $u'(z)$ and $v'(z)$ are then filtered to include only the peak vertical wave number and the adjacent wave numbers. A first guess for the associated time window, $[t_s^0, t_e^0]$, is obtained from the positions of the adjacent minima. At this point, the vertical wavelength assigned to the event is determined from the power-weighted mean wave number

of the selected time–wave number window. The initial altitude interval $[h_s^0, h_e^0]$ is $[0, 18]$ km. The degree of polarization $d$ is next computed for the initial window and all possible combinations of shorter windows contained in it. The window defined by $[t_s, t_e]$, $[h_s, h_e]$, where $t_s^0 \le t_s < t_e \le t_e^0$ and $h_s^0 \le h_s < h_e \le h_e^0$, with the greatest total number of data points and a polarization $d \ge 0.8$ is saved for subsequent analysis. If no window inside $[t_s^0, t_e^0]$, $[h_s^0, h_e^0]$ with $d \ge 0.8$ can be found, then the peak is discarded. Otherwise $\epsilon$ and $\Theta$ are computed using the Stokes parameters.

The angle $\Theta$ of the major axis of the ellipse counterclockwise from the eastward direction is the direction of wave propagation, but with an ambiguity of $180°$. To resolve this ambiguity, we also compute this angle as in Evan and Alexander (2008):

$$\Theta_{T'_{+90}} = \tan^{-1}\left(\frac{\overline{u'T'_{+90}}}{\overline{v'T'_{+90}}}\right), \tag{1}$$

where $T'_{+90}$ is the value of the temperature perturbation after shifting the phase by $+90°$ using a Hilbert transform. Finally, the ratio $\epsilon$ of the minor to the major axis is related to the intrinsic frequency $\Omega$ of the gravity wave as

$$\epsilon = \frac{f}{\Omega} - \frac{k_h m \overline{V_z}}{[m^2 + 1/(4H_s^2)]\Omega}, \tag{2}$$

where $f$ is the Coriolis parameter, $H_s$ the density scale height, $k_h$ the magnitude of the horizontal wave number vector, and $\overline{V_z} = \overline{(\mathrm{d}V/\mathrm{d}z)}$ the mean shear of the horizontal wind in the direction perpendicular to $\Theta$. To compute $\overline{V_z}$, we first average $V$ over $[h_s, h_e]$ and $[t_s, t_e]$.

The dispersion relation for inertia–gravity waves (e.g., Hankinson et al., 2014) is

$$\Omega^2 = f^2 + \frac{N^2 k_h^2}{m^2 + 1/(4H_s^2)} - \frac{2fk_h m \overline{V_z}}{m^2 + 1/(4H_s^2)}, \tag{3}$$

where $N$ is the Brunt–Väisälä frequency. Combining Eq. (2) with Eq. (3) gives

$$\Omega^2\left(\epsilon^2 - a^2\right) - 2\Omega f \epsilon\left(1 - a^2\right) + f^2\left(1 - a^2\right) = 0, \tag{4}$$

where $a^2 = \overline{V_z^2}/(N^2[1 + (4m^2 H_s^2)^{-1}])$. Solving for $\frac{\Omega}{f}$, this results in

$$\frac{\Omega}{f} = \frac{\epsilon\left(1 - a^2\right) \pm \left[\epsilon^2\left(1 - a^2\right)^2 - \left(\epsilon^2 - a^2\right)\left(1 - a^2\right)\right]^{1/2}}{\epsilon^2 - a^2}. \tag{5}$$

After determining $\Omega$, $k_h$ follows from the rearranged dispersion relation (3):

$$k_h = k_{\text{disp}} = \frac{f m \overline{V_z}}{N^2} \tag{6}$$

$$\pm \frac{1}{N}\left[\frac{f^2 m^2}{N^2}\overline{V_z}^2 + \left(\Omega^2 - f^2\right)\left(m^2 + \frac{1}{4H_s^2}\right)\right]^{1/2},$$

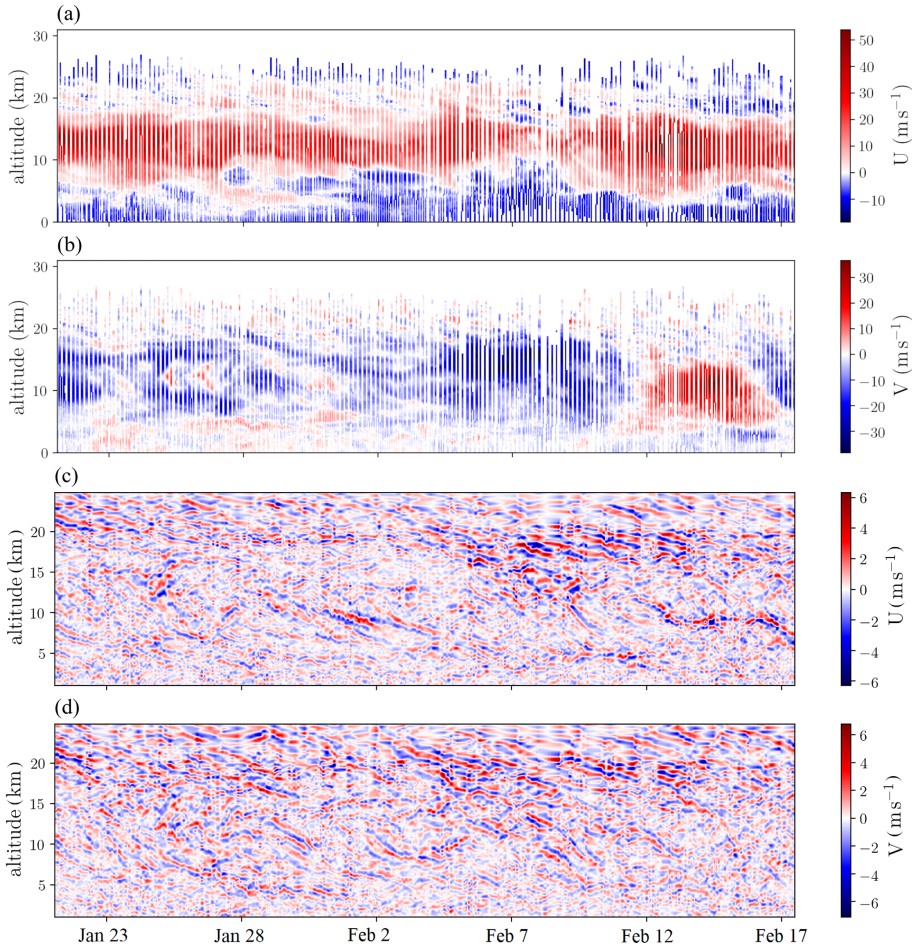

**Figure 2. (a)** Zonal and **(b)** meridional wind measured by ascending and descending soundings at the BCO. **(c)** Zonal and **(d)** meridional wind perturbations after filtering the raw measurements to $500\,\mathrm{m} < \lambda_z < 10\,\mathrm{km}$ and $3\,\mathrm{h} < \tau < 53\,\mathrm{h}$.

where by definition $k_\mathrm{h}$ is positive in the direction of wave propagation.

To obtain the ground-based frequency $\omega$, we compute at each altitude the quadrature spectrum between $u'(t)$ and $v'(t)$, $QS(t, \nu)$, using the S transformation (Stockwell et al., 1996):

$$QS(z; t, \nu) = \Im\left[ST_u(t, \nu) \cdot ST_v^*(t, \nu)\right]\big|_z, \qquad (7)$$

where $ST_u(t, \nu)$ is the S transformation in the time dimension of $u'(t)$ at a given $z$, and $\nu$ denotes frequency. Then, $\omega$ corresponds to the peak in $QS(z; t, \nu)$ after integrating over $[h_\mathrm{s}, h_\mathrm{e}]$ and $[t_\mathrm{s}, t_\mathrm{e}]$.

In principle we could estimate $\omega$ from the background wind in the direction of wave propagation, $U_\mathrm{h}$, using $\omega = \Omega + k_\mathrm{h} U_\mathrm{h}$. However, due to the strong wind shear (see Fig. 2a and b), these results would be highly sensitive to the altitude range selected to compute $U_\mathrm{h}$. As the Doppler shift equation assumes $U_\mathrm{h}(z) = \mathrm{const}$, it is not clear if $U_\mathrm{h}$ is best approximated as an average across an altitude range or the wind at a particular level. Moreover, as $U_\mathrm{h}$ is the projection of the

horizontal wind along the wave propagation direction, an additional uncertainty is introduced by the uncertainty in $\Theta$. For this reason we compute $\omega$ with the alternative method described above.

### 2.4 Normal-mode decomposition of numerical data

We derive global wave spectra as well as vertical profiles of regionally averaged divergence and vorticity from the ERA5 reanalysis and a numerical simulation. For both data sets we analyze 3-hourly horizontal winds between 20 January–29 February 2020, matching the EUREC[4]A period. The hodograph analysis is not applied to the numerical data, because we found the vertical resolution to be too coarse.

ERA5 is produced and made publicly available by the European Centre for Medium Range Weather Forecasts (ECMWF). The original data with a horizontal resolution of $\sim 30\,\mathrm{km}$ are stored on 137 hybrid sigma/pressure levels from the surface up to an altitude of 80 km (0.01 hPa).

The numerical simulation run at 2.5 km horizontal resolution is performed with the ICON model (Zängl et al., 2015).

https://doi.org/10.5194/wcd-3-1-2021

It was initialized at 00:00 UTC on 20 January 2020 from an operational analysis of the atmospheric state produced by the ECMWF and forced at the lower boundary with 5 d CE1 running-mean sea surface temperatures, which are also taken

from an ECMWF analysis. Ninety vertical levels extend from the surface up to 75 km with a Klemp-type implicit Rayleigh damping (Klemp et al., 2008) of vertical winds starting at 44 km. The simulation is freely evolving and integrated until 00:00 UTC on 1 March 2020.

We re-grid the data from both ERA5 and ICON to a regular n256 Gaussian grid with $1024 \times 512$ points in longitude and latitude, respectively, corresponding to a resolution of 39 km at the Equator. In the vertical we interpolate to 68 hybrid sigma/pressure levels between the surface and 10 hPa

($\sim$ 31 km).

The data at each time step are then subjected to a normal-mode function decomposition, using the freely available software package MODES, a detailed description of which is given in Žagar et al. (2015). The modal decomposi-

20 tion projects the three-dimensional fields of geopotential height and horizontal wind onto an orthogonal set of basis functions. These functions are solutions to the Rossby and inertia–gravity wave (IG) dispersion relationships, respectively (Kasahara, 1976). Since the former are vorticity-

25 dominated, we label them as ROT. The orthogonality of the basis functions allows filtering out specific wave modes, including the option of a subsequent inversion back to physical space. This allows us to derive the local wind fields inside the EUREC[4]A domain, as well as corresponding divergence

and vorticity fields, associated with specific modes.

## 3 Results

### 3.1 Effects of the global wave spectrum on mesoscale divergence

Figure 3a and b show how vertical profiles of area-averaged

divergence vary in time. The strong vertical variability in the profiles derived from the array of all soundings and the circle, respectively, is consistent with previous observations by Bony and Stevens (2019) and Stephan et al. (2020). Vorticity varies on longer scales in time–altitude space (Fig. 3c and

40 d). For ERA5 and ICON we display the corresponding divergence and vorticity profiles associated with IG and ROT modes, respectively (Fig. 4). These profiles are computed for the same geographical region and an equivalent radius of $\sim$ 260 km. This radius corresponds to the smallest radius

that allows us to compute a centered area average, i.e., using odd numbers of grid points in each direction on the n256 Gaussian grid, while at the same time exceeding the truncation scales applied in the normal-mode decomposition in both horizontal directions (truncation meridional wave num-

ber $n = 200$; zonal wave number $k = 320$). Divergence associated with ROT modes is negligible; ROT modes possess only 2.4 % (ERA5) and 3.3 % (ICON) of the variance contained in the IG modes of Fig. 4a and b. Similarly, IG modes possess only 3.8 % (ERA5) and 3.1 % (ICON) of the variance contained in the ROT modes of Fig. 4c and d. Figure 4 also

shows strong variability at short scales in divergence and at long scales in vorticity, with similar magnitudes to those observed. Notably, 3D-Var, which ingests a 3 h running mean of measurements, and MODES, which is independently applied at single time steps, yield divergence as well as vor-

ticity anomalies that exhibit temporal persistence with vertically coherent structures. From the comparison between observations and numerical data we conclude that by far the largest fraction of divergence variability inside the sounding network is due to IG waves.

Because the EUREC[4]A sounding network sampled areas of different sizes, we can for the first time address how observed magnitudes of area-averaged divergence variability change with area. Figure 5 shows scatterplots of divergence amplitudes multiplied by the corresponding area-equivalent

radius versus the equivalent radius for three different tropospheric layers. We here define divergence amplitude as the difference between the maximum and minimum value found in a layer. The running means (thick blue dots) follow lines that are approximately horizontal, albeit with a slight upward

slope at $h = 6$–9 km. Thus, divergence amplitudes at equivalent radii of 100–300 km scale approximately inversely with radius.

ERA5 and ICON also show such an inverse scaling of divergence amplitudes (Fig. 6). As expected from the predom-

80 inant IG contribution to divergence, this scaling is due to the scaling of the IG-driven divergence field. While the lower bound in $R$ is limited by our choice of normal-mode expansion parameters, we can compute divergence for areas that exceed the sizes sampled during EUREC[4]A. Figure 6 shows

that the approximate $R^{-1}$ scaling of divergence amplitudes remains valid across the entire mesoscale, up to at least the synoptic scale of about 1000 km (zonal wave number 38). ERA5 and ICON produce qualitatively similar results, but we notice in Fig. 6 that ERA5 has slightly larger divergence

amplitudes than ICON. To shed light on this difference and, more importantly, to find an explanation for the scaling with radius, Fig. 7 examines the global energy spectra in ERA5 and ICON and their decomposition into IG and ROT modes.

In both data sets the total energy follows a $k^{-1}$ power

law at synoptic scales that transitions to a steeper slope at $k \approx 8$. For $k > 8$ the ROT spectrum in ERA5 scales as $k^{-3}$, and the IG spectrum follows on average a $k^{-5/3}$ slope but with substantial deviations from a constant slope. Towards the largest $k$ there is a further steepening in ERA5, which

was also reported for ERA-Interim and points to a lack of energy at small scales in the reanalysis (Žagar et al., 2015). The spectral behavior in ICON is similar to ERA5, but the slopes for different regimes are more sharply defined. The IG curve follows a $k^{-1/2}$ slope for $k < 8$ and then $k^{-5/3}$

up to $k \approx 100$. At $k > 100$, there is a slight steepening in

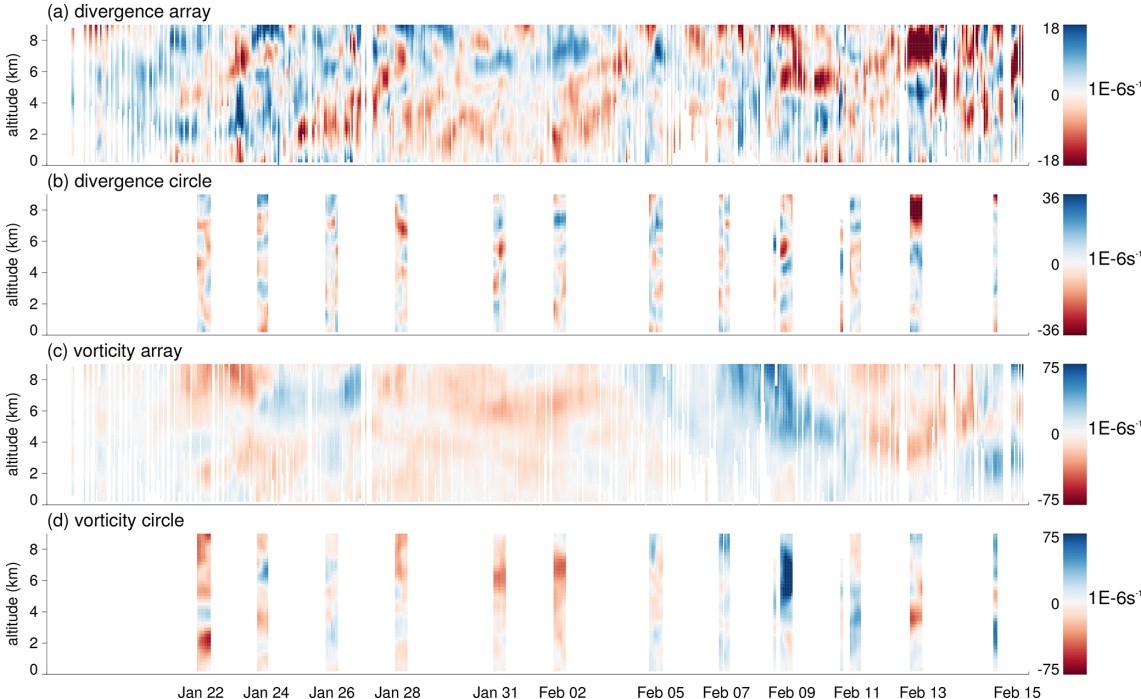

**Figure 3. (a, b)** Divergence and **(c, d)** vorticity anomalies derived from 3D-Var applied to **(a, c)** the sounding network and **(b, d)** the HALO circles.

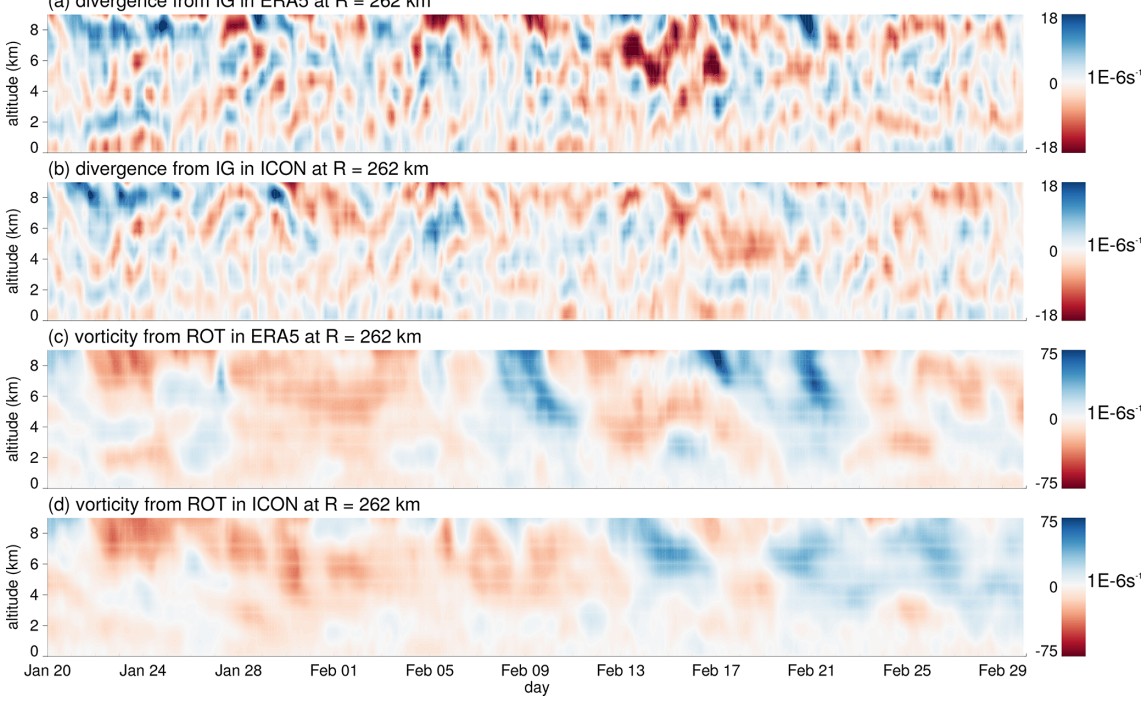

**Figure 4. (a, b)** Divergence and **(c, d)** vorticity anomalies associated with **(a, b)** IG and **(c, d)** ROT modes in **(a, c)** ERA5 and **(b, d)** ICON for an equivalent radius of 262 km. The area is centered on $\sim$ 12° N, $-57$° E, matching the approximate location of the EUREC[4]A sounding network (compare Fig. 1b).

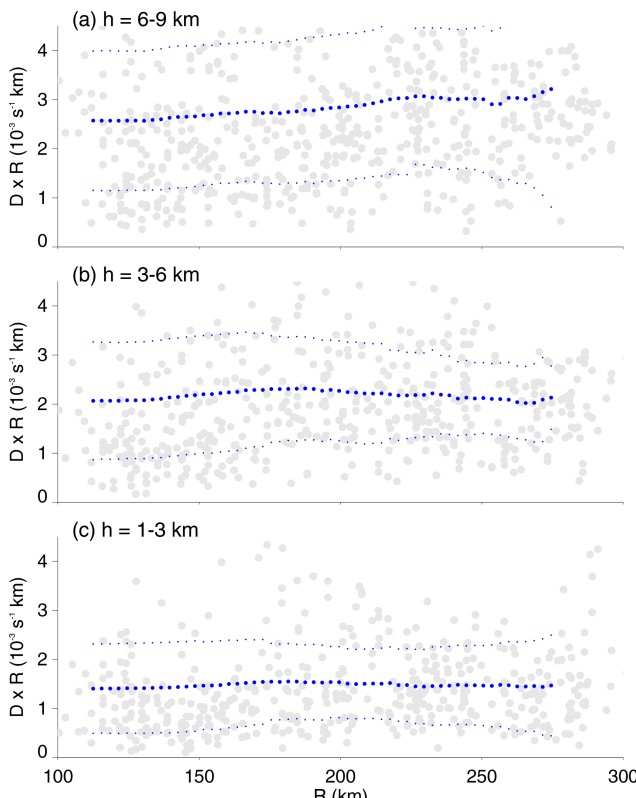

**Figure 5.** Divergence amplitude anomaly (defined as maximum minus minimum at **(a)** 6–9 km, **(b)** 3–6 km, and **(c)** 1–3 km altitude) multiplied by equivalent radius, as a function of equivalent radius. Based on divergence from 3D-Var applied to the sounding network. Thick (thin) blue dots show the running mean ($\pm 1$ standard deviation), using $R$ intervals of 25 km.

ICON as well, such that the slope between $k \approx 100$–150 is $k^{-2}$. The smallest scale shown in Fig. 6 is $k = 153$. The three prominent slopes of $k^{-1/2}$, $k^{-5/3}$, and $k^{-2}$ are marked with magenta lines in Fig. 7b. Figure 8 shows the energy contained in IG modes in ERA5 relative to ICON. For $k < 100$ the energy in ERA5 exceeds the energy in ICON, and vice versa for $k > 100$. Most of the divergence variability considered here can be attributed to the more energetic large scale, i.e., $k < 100$, explaining the larger divergence amplitudes in ERA5.

As it turns out, the spectral slopes can be used to predict the magenta lines in Fig. 6b; i.e., they can explain how divergence magnitudes vary with equivalent radius. Consider a gravity wave with wave vector $(k, l, m)$, frequency $\omega$, and phase $\phi$. The associated wind perturbations can be expressed as

$$(u, v, w) = (A_x, A_y, A_z)\sin(kx + ly + mz - \omega t + \phi), \quad (8)$$

where the amplitudes $A_i$ are functions of the wave vector. The associated horizontal divergence scales with $k$ and $l$ as

follows:

$$\mathrm{Div} \sim (kA_x + lA_y)\cos(kx + ly + mz - \omega t + \phi). \quad (9)$$

As we will demonstrate in Sect. 3.2, the IG waves inside the EUREC⁴A domain were mainly propagating zonally, so that we can assume divergence to scale as $\mathrm{Div} \sim kA_x(k)$. Near-zonal propagation is also expected, because IG waves can become equatorially trapped (Wheeler and Kiladis, 1999), and it is these equatorially trapped east- or westward-propagating modes that we identify with the normal-mode decomposition. How $A_x(k)$ depends on $k$ is contained in the identified spectral slopes of IG modes in Fig. 7b: $A_x$ scales as $k^{-1/4}$ for $k < 8$, as $k^{-5/6}$ for $k = 8$–100, and as $k^{-1}$ for $k = 100$–150. To derive the relationship between these spectral slopes and the divergence amplitudes, we follow the ansatz that each averaging scale $R^* = 2\pi/k^*$ feels the divergence on all scales $R > R^*$, whereas divergence at scales $R < R^*$ has little or no effect. Under this assumption, the scaling of divergence amplitudes $A_D(k^*)$ can be approximated by the sum over all $k$ smaller or equal to $k^* = 2\pi/R^*$:

$$A_D(k^*) \sim \sum_{k=1}^{k=k^*} k \cdot A_x(k) \sim \sum_{k=1}^{k=k^*} k \cdot k^{\sigma(k)}, \quad (10)$$

with $\sigma(k) \in \{-1/4, -5/6, -1\}$, as indicated above. Computing this sum and multiplying by a constant to match the IG point at $R = 1044$ km in Fig. 6b, d, and f results in the solid magenta lines drawn in these panels. Evidently, at 3–6 and 6–9 km altitude, the match is excellent. At 1–3 km Fig. 6f misses the decrease in divergence magnitude at small $R$. This might be due to an introduction of additional energy at short horizontal scales by convection, which tends to have tops at 1–2 km during EUREC⁴A. We did not perform any calculation for ERA5, as ERA5 clearly lacks energy at small scales and the ICON simulation is more realistic in this regard. However, as noted above, even in ICON the slope of the IG spectrum steepens slightly at $k = 100$. Since it is not clear if the change in spectral slope from $k^{-5/6}$ to $k^{-2}$ at $k = 100$ is physical, we repeat the calculation using a slope of $k^{-5/6}$ for all $k \geq 8$. This results in the dashed magenta lines of Fig. 6b, d, and f. Now, for $R < 392$ km there is no longer a drop in amplitude, but amplitudes remain approximately constant. This behavior better matches the observations (Fig. 5). The validity of Eq. (10) is further discussed in Sect. 4.

## 3.2 Effects of individual waves on mesoscale divergence

In the previous section we related the characteristics of the global IG spectrum to the statistical properties of variability in area-averaged mesoscale divergence. We can now explain divergence magnitudes in a statistical sense, but the longevity of structures identified in Figs. 3 and 4 suggests that individual waves may dominate the local divergence field at many times. It is important to note that the result obtained in the

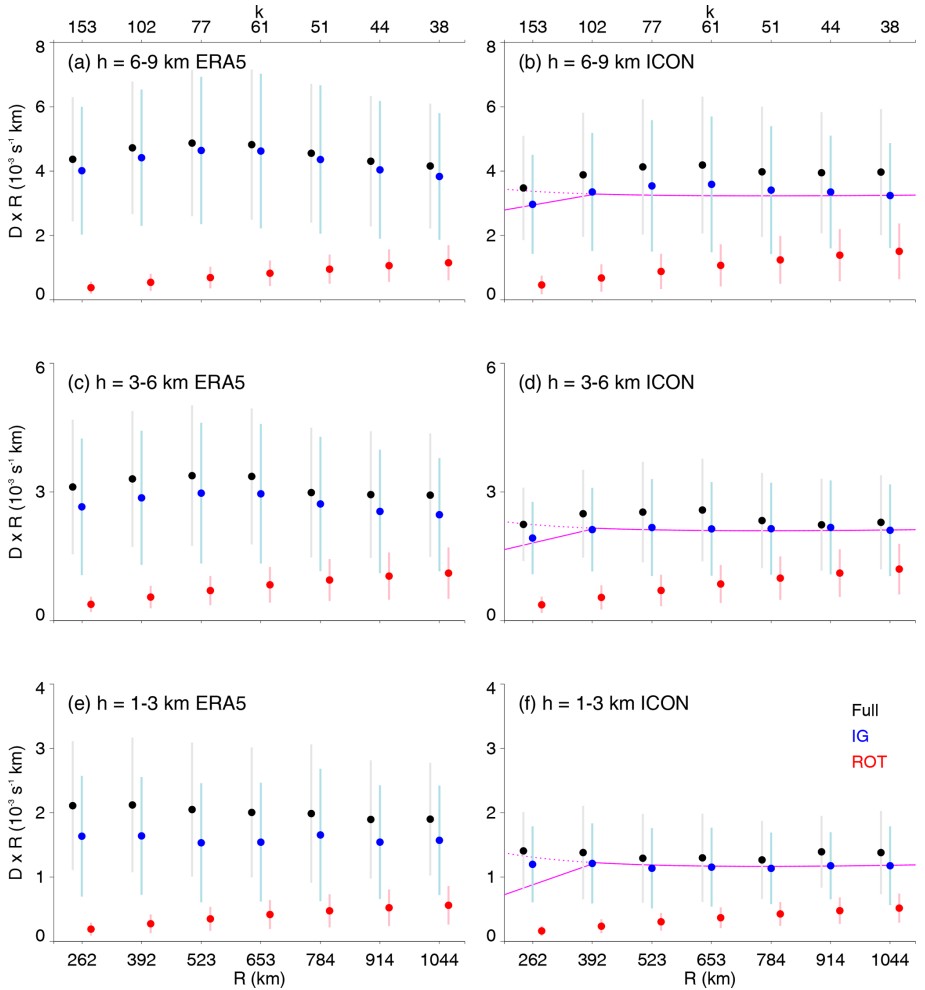

**Figure 6.** Divergence amplitude anomaly (defined as maximum minus minimum at **(a)** 6–9 km, **(b)** 3–6 km, and **(c)** 1–3 km altitude) in **(a, c, d)** ERA5 and **(b, d, f)** ICON. The vertical lines mark ±1 standard deviation. The solid magenta lines in panels **(b, d, f)** are theoretical predictions based on the spectra shown in Fig. 7 (see the text for details). The dashed magenta line uses a spectral slope of $k^{-5/3}$ instead of $k^{-2}$ at $k > 100$.

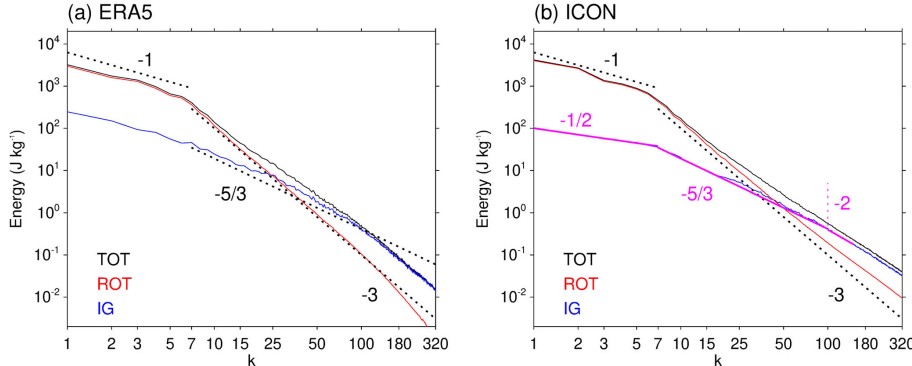

**Figure 7. (a, b)** Energy spectra as a function of zonal wave number from MODES applied to **(a)** ERA5 and **(b)** ICON. The (black) total energy spectrum is decomposed into (red) ROT modes and (blue) IG modes. Dashed black lines show spectral slopes of $k^{-1}$, $k^{-5/3}$, and $k^{-3}$; magenta lines in panel **(b)** show $k^{-1/2}$, $k^{-5/3}$, and $k^{-2}$, where the transition between $k^{-5/3}$ and $k^{-2}$ is at $k = 100$ and marked by a thin vertical magenta line.

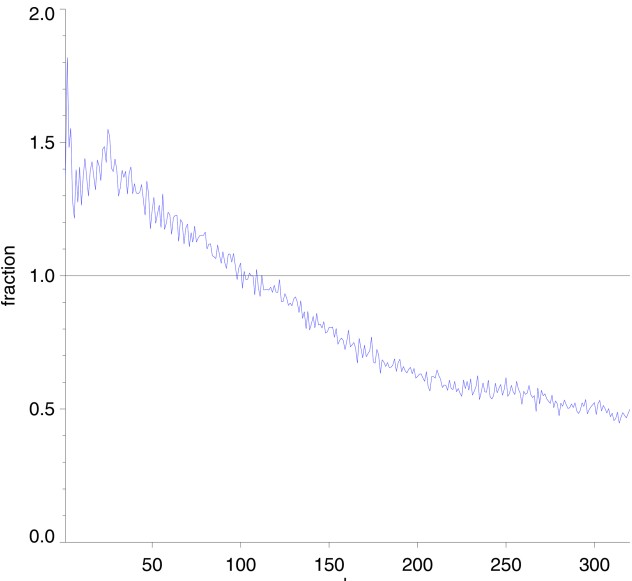

**Figure 8.** Fraction of the energy in IG modes in ERA5 relative to ICON as a function of zonal wave number (ERA5 divided by ICON).

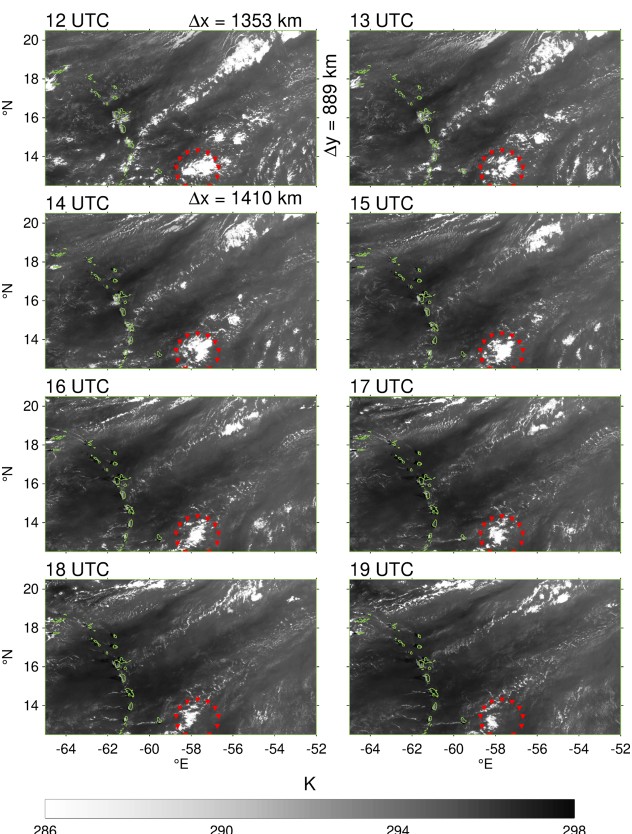

**Figure 9.** GOES-16 images from 31 January 2020, on a regular longitude–latitude grid. The top edge of the images measures 1358 km, the bottom edge 1410 km, and the sides 889 km. Shown are brightness temperatures from the C13 channel. Green outlines show the Caribbean Islands, and red triangles the locations of HALO's dropsondes.

previous section does not require a saturated wave spectrum. Evidently, the local spatiotemporal predominance of one or a few waves would change the scaling law in the given period and region, but this is consistent with the relatively large standard deviations in Figs. 5 and 6. Therefore, we next investigate if the generic IG waves that appear to govern the statistics of mesoscale divergence can directly modulate convection through their individual impact on mesoscale vertical motion.

To study local wave characteristics, we use the iterative hodograph algorithm, which is designed specifically to detect nearly monochromatic waves in soundings (see Sect. 2.3). The algorithm extracted 34 wave events with polarization $d \geq 0.8$ and with an average polarization of 0.84. All events and extracted wave characteristics are listed in Table 1. Most detected waves have vertical wavelengths of 1.2–4 km with a mean of 2.6 km and highly variable horizontal wavelengths with a mean and standard deviation of $\mathcal{O}(450)$ km. Intrinsic periods range from 6–45 h with a mean of 21 h, and ground-based periods from 11–50 h with a mean of 28 h. The average propagation direction, computed with two different methods, is eastward.

The identified wave characteristics do not show any evidence for systematic changes over the period of the campaign. This is consistent with a generally broad but unsaturated tropospheric IG background. If the wave background were saturated, then it would be impossible to identify waves with a high degree of polarization. Instead, at most times, we are able to find predominant waves (Table 1).

A day on which gravity waves may have had an effect on the cloud field is 31 January. Figure 9 shows hourly infrared satellite images of the campaign region on this day. A smooth modulation of the cloud field is clearly visible in the form of alternating southwest-to-northeast-oriented bands of dry and cloudy areas. The crests of the cloudy areas are topped by narrow stripes of bright clouds. One of the larger cloud clusters seen in the images is located inside the HALO circle. The top and bottom height of this cloud are $\sim 2.3$ and $\sim 1$ km, respectively. In the cloud layer the mean zonal wind is $-4.3\,\mathrm{m\,s^{-1}}$ and the mean meridional wind is $-0.1\,\mathrm{m\,s^{-1}}$. Thus, the motion of the cloud cluster across the circle (diameter of 200 km) matches the estimated advection of 100 km in 6.5 h.

The 31st of January is also the day when the wave algorithm detected the most distinct wave event. Three out of the four data sets from the surface-based stations contain a wave signal with a high degree of polarization (0.86, 0.86, 0.87; Table 1). Notably, the three detected waves agree broadly in terms of the estimated horizontal wavelengths (480, 340, 440 km), which match well the 400 km spacing of the respective moist or dry regions in Fig. 9. In addition, the long

**Table 1.** For all identified wave events, columns list the (1) time interval $t_1$–$t_2$ as "start date/start hour–end date/end hour" with hour in UTC, (2) data set **S** (R: *RH-Brown*; M: *Meteor*; B: BCO; C: combined), (3) altitude interval $h_1$–$h_2$ used for the computation of Stokes parameters, (4) degree of polarization $d$, (5) wave propagation direction $\Theta$ counterclockwise from the eastward direction computed with the Stokes parameters, (6) wave propagation direction $\Theta_{T_{90}}$ obtained with Eq. (1), (7) vertical wavelength $\lambda_z$, (8) horizontal wavelength $\lambda_h$, (9) intrinsic period $\tau^*$, and (10) ground-based period $\tau$ obtained from the quadrature spectrum.

| $t_1$–$t_2$ day/hour | S | $h_1$–$h_2$ m | $d$ | $\Theta$ ° | $\Theta_{T_{90}}$ ° | $\lambda_z$ m | $\lambda_h$ km | $\tau^*$ h | $\tau$ h |
|---|---|---|---|---|---|---|---|---|---|
| **January** | | | | | | | | | |
| 14/01–14/11 | R | 6000–15710 | 0.84 | 15 | 3 | 3306 | 718 | 29 | 19 |
| 18/15–19/00 | R | 4000–15710 | 0.81 | −44 | −3 | 2198 | 521 | 30 | 15 |
| 20/21–21/07 | R | 6000–13710 | 0.80 | 17 | −3 | 1585 | 274 | 24 | 11 |
| 21/15–22/07 | B | 5030–15810 | 0.86 | 25 | −9 | 5365 | 1836 | 37 | 21 |
| 23/23–24/17 | M | 6000–11750 | 0.86 | 34 | −20 | 1043 | 103 | 14 | 27 |
| 25/05–25/19 | M | 6000–17750 | 0.80 | 4 | −1 | 5190 | 226 | 7 | 25 |
| 25/08–26/10 | B | 13020–17810 | 0.80 | 18 | −23 | 1848 | 234 | 19 | 41 |
| 26/04–26/16 | C | 6000–17640 | 0.82 | 2 | 1 | 2502 | 668 | 33 | 19 |
| 27/02–27/17 | B | 1030–13810 | 0.86 | −13 | 4 | 4847 | 430 | 12 | 41 |
| 27/10–27/19 | C | 4000–15640 | 0.81 | 32 | −10 | 1937 | 159 | 12 | 33 |
| 27/21–28/11 | M | 4000–15750 | 0.85 | −27 | 9 | 2845 | 1561 | 45 | 22 |
| 28/07–29/02 | C | 0–17640 | 0.81 | −16 | 20 | 3553 | 126 | 6 | 26 |
| 29/20–30/05 | B | 5030–15810 | 0.86 | −4 | 2 | 2861 | 1100 | 39 | 19 |
| 29/21–31/02 | R | 8000–17710 | 0.81 | −4 | 4 | 2526 | 861 | 38 | 33 |
| 31/14–01/00 | C | 3000–8640 | 0.85 | −25 | −2 | 1749 | 175 | 14 | 50 |
| 30/16–31/05 | C | 10000–17640 | 0.82 | 16 | 0 | 1596 | 174 | 17 | 50 |
| 31/05–01/00 | B | 7030–13810 | 0.86 | −14 | −3 | 3192 | 477 | 22 | 33 |
| 31/14–31/23 | M | 4000–7750 | 0.86 | −28 | −1 | 2286 | 339 | 20 | 27 |
| 31/17–01/02 | R | 5000–9710 | 0.87 | −11 | 4 | 3962 | 441 | 16 | 34 |
| **February** | | | | | | | | | |
| 01/01–01/16 | M | 6000–13750 | 0.85 | 0 | −0.63 | 1302 | 120 | 14 | 19 |
| 01/22–02/12 | B | 3030–13810 | 0.86 | 20 | −15 | 1279 | 68 | 8 | 36 |
| 02/20–03/07 | C | 5000–11000 | 0.85 | −27 | 10 | 1944 | 138 | 10 | 41 |
| 01/07–02/06 | R | 4000–13710 | 0.80 | −24 | −7 | 2987 | 536 | 24 | 31 |
| 05/03–05/15 | C | 4000–11000 | 0.85 | −15 | −5 | 993 | 179 | 24 | 39 |
| 05/11–05/20 | R | 1000–10710 | 0.86 | −33 | −3 | 815 | 64 | 12 | 44 |
| 05/14–06/11 | B | 4030–7000 | 0.86 | 22 | 7 | 1062 | 191 | 23 | 22 |
| 05/12–05/22 | M | 0–3750 | 0.86 | −12 | −4 | 2435 | 111 | 8 | 16 |
| 07/05–07/17 | C | 0–17640 | 0.80 | −41 | 10 | 1882 | 80 | 7 | 18 |
| 07/18–08/08 | R | 0–11710 | 0.80 | 28 | 6 | 4264 | 565 | 21 | 29 |
| 08/08–09/03 | B | 903–15810 | 0.85 | 0 | 5 | 6964 | 1395 | 27 | 33 |
| 08/21–09/09 | R | 6000–15710 | 0.82 | 21 | −4 | 2895 | 417 | 21 | 15 |
| 09/13–10/04 | B | 5030–11810 | 0.85 | −30 | −7 | 2776 | 638 | 29 | 29 |
| 11/09–11/21 | M | 6000–13750 | 0.85 | −41 | 0 | 775 | 41 | 8 | 12 |
| 15/17–16/07 | B | 13030–17810 | 0.86 | 27 | 9 | 3179 | 711 | 28 | 31 |
| **Average** | | | 0.84 | −3 | 0 | 2645 | 461 | 21 | 28 |
| **SD** | | | 0.02 | 23 | 9 | 1434 | 445 | 10 | 11 |

ground-based periods (33, 27, 34 h) are consistent with the nearly stationary appearance of the cloud field in Fig. 9.

The propagation direction indicated by the algorithm, however, is more due eastward (−14°, −28°, −11°) than one would expect from the alignment of the cloud pattern. It could be that vertical propagation and refraction of the wave are partly responsible for this difference, because the ships were located further to the south, where a patterning of clouds is not visible (not shown). To obtain an additional estimate of wave parameters that allows for a more direct comparison with Fig. 9, we apply a hodograph analysis to the 66 HALO dropsonde profiles collected on 31 January over the course of multiple circle flights. Their locations are marked by red triangles in Fig. 9. We pre-select the altitude inter-

val of 2–5 km in order to include the upper portion of the cloud layer while avoiding noise near the surface. Since the entire duration of the flight is less than 8 h, we compute wind perturbations by removing the average of all profiles instead of filtering in time. The resulting horizontal wavelength is 421 km with a propagation angle $\Theta = -43.1°$, in agreement with Fig. 9.

The good match of the satellite image with wave properties found independently in data from four stations is a strong indication for a causal link. The 31st of January was the only day that showed a potential wave imprint on the cloud field that was detectable by eye. Thus, we conclude that visible effects of individual waves on convection occur rather seldom, at least during boreal winter in the studied region.

## 4    Conclusions

This study extended the analysis of recent novel measurements of mesoscale divergence variability (Bony and Stevens, 2019) by diagnosing vertical profiles of mesoscale divergence in the extensive sounding network of the 2020 EUREC[4]A field campaign. For the first time, we could measure how divergence magnitudes depend on the area over which averages are computed, which may be relevant to sub-daily variability of cloudiness (Mauger and Norris, 2010; Vogel et al., 2020; George et al., 2020). Observed divergence magnitudes during EUREC[4]A scale approximately inversely with the area-equivalent radius, a result that is also true for the ERA5 reanalysis and a 2.5 km global numerical simulation. By applying a normal-mode decomposition to the numerical data, we demonstrated that the major fraction of mesoscale divergence variability is due to inertia–gravity (IG) waves, confirming the hypothesis put forward by Stephan et al. (2020).

Moreover, we derived an analytic relationship between the wave number dependence of global IG energy spectra and the scaling of divergence amplitudes with area. The essential ansatz for this Eq. (10) is that IG waves on all scales larger than a considered area contribute to the divergence averaged over this area. The data presented in this study follow Eq. (10) very closely. Yet the relationship is not trivial from an intellectual point of view, as one might expect some cancelation of the local divergence induced by some IG waves with the local convergence associated with other IG waves. Either this cancelation has no effect on the overall scaling of divergence amplitudes (note that Eq. 10 only predicts their scaling with area, not their absolute magnitude) or the cancellation is negligible. The second possibility is more likely to be true if the tropospheric IG wave background is not saturated but locally composed of few waves. Persistent features in filtered-wind and divergence profiles suggest that the latter possibility may apply.

To gain more information about the properties of local waves, we extracted nearly monochromatic wave events from soundings by means of an iterative hodograph analysis. For most days, clear IG signals could be found, and on one day the identified wave likely had a direct effect on the organization of clouds, as evidenced by comparison to satellite images. Thus, although seldom, direct wave imprints on convection do occur. This suggests that the IG wave background is indeed not saturated. However, the systematic modulation of vertical motion due to IG waves of different scales may be more relevant for convection. More research is required to quantify the relative importance of the stochastic wave forcing for shaping patterns of convection, compared to other well-known local influences, such as lower-tropospheric stability (Medeiros and Stevens, 2011), relative humidity (Slingo, 1987), and sea surface temperature (Qu et al., 2015).

Future studies should repeat the measurements analyzed in this study for different regions, especially different latitudes, as well as regions over land, to test the universal applicability of the proposed relationship between energy spectra and divergence amplitudes. If confirmed, then our results imply that all scales are relevant for mesoscale variability, which has consequences for regional models that introduce an artificial truncation scale, and could be used to design better external forcing methods for regional models.

*Code availability.* MODES software can be obtained at https://modes.cen.uni-hamburg.de/download (last access: 25 October 2020). ERA5 is produced and made publicly available by the European Centre for Medium Range Weather Forecasts (ECMWF).

*Data availability.* The vertically gridded (Level 2) radiosounding data described by Stephan et al. (2021) in NetCDF format are available to the public at https://doi.org/10.25326/62 TS1. GOES-16 Advanced Baseline Imager data are provided by NASA's Worldview application (https://worldview.earthdata.nasa.gov, last access: 24 November 2020). Data from the numerical simulation are stored by the Excellence in Science of Weather and Climate in Europe (ESiWACE) project at the German Climate Computing Center (DKRZ).

*Author contributions.* CCS conceptualized the study and prepared the manuscript. AM carried out the hodograph analysis and provided Fig. 2.

*Competing interests.* The authors declare that they have no conflict of interest.

*Acknowledgements.* Claudia Christine Stephan was supported by the Minerva Fast Track Programme of the Max Planck Society. She would like to thank David Raymond for his help with the 3D-Var software, which is openly available for download at http://kestrel.nmt.edu/~raymond/software/candis/candis.html (last ac-

*Please note the remarks at the end of the manuscript.*

cess: 21 October 2019). She is also grateful to Nedjeljka Žagar for her advice on using the MODES software.

*Financial support.* This research has been supported by the Max-Planck-Gesellschaft.

*Review statement.* This paper was edited by Pedram Hassanzadeh and reviewed by two anonymous referees.

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
