# Peer review of "The signature of the tropospheric gravity wave background in observed mesoscale motion"

_Weather and Climate Dynamics, 2020_

## Referee Comment (RC1) · Anonymous Referee #1 · 2 Feb 2021

This paper studies the gravity wave characters through an extensive sounding network of a recent campaign and high-resolution simulation of the same period. A novel analysis is performed to connect the energy spectra of the inertial gravity waves and the observed scaling of divergence amplitudes with area. This manuscript is well written and easy to follow. The presentation is very clear, and the scientific approach is also valid. Yet, a few interesting points were raised while reviewing this manuscript. Therefore, a minor revision is suggested here before final publication of the paper.

Specific Comments:

1. decomposition of numerical data:

[Figure]

While analyzing the observation data, a Lanczos filter is used to filter out the waves with vertical wavelength between 500m and 10 km. Is similar vertical selection applied for the numerical data? How it might affect the results?

2. global spectrum

a. given that there is a lack of energy at small scales in the reanalysis data, it is not clear to what extent the steepening at k>100 in the model is realistic. Some models show a -5/3 slope till the diffusion zone. The reviewer is wondering how will it affect the calculation if the -5/3 slope holds for all the k that is greater than 8.

b. A significant amount of variation could be found in the wave amplitudes in the observation (Fig. 2cd). Is this temporal variation also holds for the global spectrum? If not, how should we reconcile this result with the direct attribution of global spectrum to the IGs?

3. Line 295: the reviewer is not fully convinced that there is a causal link between the gravity waves and the convection for the Jan 31st event. Is it possible that the good match between the wave characters and the cloud field is due to that the convection there modulates the wave signal? Also, just out of curiosity, is there a way to detect similar wave effects on the convection as this Jan 31 event in the model, especially for the 2.5km ICON run?

5. Table 1 shows a nice gravity waves analysis for the sounding data. It is interesting to know whether similar analysis could be also done for the numerical data. Moreover, the results in the models might also have implication for the sensitivity on the model resolution.

Technical Corrections:

Line 80: Missing words after "data"

Line 114: should be > 2

[Figure]

Line 235: Fig.6 -> fig.7
* * *

---

## Referee Comment (RC2) · Anonymous Referee #2 · 2 Feb 2021

This is an interesting and well written manuscript. The authors use observations collected by a sounding network to diagnose profiles of area-averaged mesoscale divergence and vorticity. Particularly, they show that the observed divergence magnitudes scale approximately inversely with the area equivalent radius, a relationship which is also confirmed in ERA5 reanalysis, and in a numerical simulation. Based on a series of assumptions, it is shown that in the numerical data, the energy spectra of inertia gravity (IG) waves might explain this scaling relationship. This paper is relevant and important for research on tropospheric mesoscale divergence and gravity waves, and their possible relationships and it fits for WCD. I have a few general points and some minor revisions to suggest.

[Figure]

General comments:

- The two most interesting findings of the paper are 1) the scaling of divergence amplitude with area equivalent radius, and 2) the possible explanation of this scaling by the wavenumber dependent IG energy spectra. While the former is robust and is confirmed in observations, reanalysis, and numerical simulation, the latter is based on a series of assumptions and is shown to be held only in the numerical data. I expect the authors to make this point clear in the text and, in particular, justify why this relationship does not hold in observations and reanalysis data.

- In the energy spectra of the numerical simulation, fitting a line with a slope of -2 on a very short segment of the spectral line (i.e., k=100-150) is not convincing, and is not justified in the text.

- Among the assumptions that have been made to derive the analytic relationship between energy spectra and divergence amplitudes, I am less convinced that IG waves should mainly propagate zonally. The authors show in section 3.2 that these waves are on average propagating eastward, however, it is also shown that there could be up to $30°$ difference between their results from the ships and HALO (why?).

- The authors propose that they will use the global energy spectra to explain why ERA5 has larger divergence amplitudes than ICON but it is not addressed in the paper. ERA5 also shows much larger variability in divergence amplitude at a given equivalent radius (Fig. 6).

Specific comments:

Line 80: data the surface -> data "cover" the surface

Line 114: It should be L/R > 2.

Line 220: thick dots -> thick blue dots

Line 237: I think you mean Fig. 7b instead of Fig. 6b.

Line 245: Add "of IG modes" after spectral slopes to make it clear.

Figure 6: Explain the magenta lines in the caption of the figure. I think it would be great if you could add a secondary x axis to this figure which shows zonal wavenumber. I assume part of the magenta line is based on $\sigma(k)$=-5/6, and part of it based on $\sigma(k)$=-1. Perhaps you could make it clear by using different color/line style.

Table 1: Please either add the column number to the table or add the column symbol to the caption.

---

## Author Comment (AC1) · 23 Feb 2021

Authors' response to reviewers' comments

Reviewer #1 This paper studies the gravity wave characters through an extensive sounding network of a recent campaign and high-resolution simulation of the same period. A novel analysis is performed to connect the energy spectra of the inertial gravity waves and the observed scaling of divergence amplitudes with area. This manuscript is well written and easy to follow. The presentation is very clear, and the scientific approach is also valid. Yet, a few interesting points were raised while reviewing this manuscript. Therefore, a minor revision is suggested here before final publication of

the paper.

We thank the reviewer for these encouraging remarks and valuable comments.

Specific Comments: 1. decomposition of numerical data: While analyzing the observation data, a Lanczos filter is used to filter out the waves with vertical wavelength between 500m and 10 km. Is similar vertical selection applied for the numerical data? How it might affect the results?

The Lanczos filter is only used for the hodograph analysis and the hodograph analysis is only applied to observations, not to numerical data. When we compare observations to numerical data, no filtering is used. To avoid any misunderstanding we added the word "unfiltered" at line 98: "The computation of divergence and vorticity from unfiltered sounding data. . ."

2. global spectrum

a. given that there is a lack of energy at small scales in the reanalysis data, it is not clear to what extent the steepening at k>100 in the model is realistic. Some models show a -5/3 slope till the diffusion zone. The reviewer is wondering how will it affect the calculation if the -5/3 slope holds for all the k that is greater than 8.

This is a very good idea. We added this analysis, which is described at lines 261-265: "Since it is not clear if the change in spectral slope from k-5/6 to k-2 at k=100 is physical, we repeat the calculation using a slope of k-5/6 for all k>8. This results in the dashed magenta lines of Fig. 6b,d,f. Now, for R<392 km there is no longer a drop in amplitude, but amplitudes remain approximately constant. This behavior better matches the observations (Fig. 5)."

b. A significant amount of variation could be found in the wave amplitudes in the observation (Fig. 2cd). Is this temporal variation also holds for the global spectrum? If not, how should we reconcile this result with the direct attribution of global spectrum to the IGs?
Please note that a global spectrum cannot be interpreted in local physical space. The same spectrum does not mean the same wind at the same location. Hence, this is an ill-posed question.

3. Line 295: the reviewer is not fully convinced that there is a causal link between the gravity waves and the convection for the Jan 31st event. Is it possible that the good match between the wave characters and the cloud field is due to that the convection there modulates the wave signal? Also, just out of curiosity, is there a way to detect similar wave effects on the convection as this Jan 31 event in the model, especially for the 2.5km ICON run?

It is well known that shallow clouds cannot generate such waves. Nevertheless, we agree that there is no definite proof for a causal link. We changed at line 286: A day on which gravity waves "most likely" -> "may have" had an effect on the cloud field

Comparing the modelled cloud field is impossible, as the details of the meteorology differ in a freely running simulation.

[note: there was no point 4. We did not omit it.]

5. Table 1 shows a nice gravity waves analysis for the sounding data. It is interesting to know whether similar analysis could be also done for the numerical data. Moreover, the results in the models might also have implication for the sensitivity on the model resolution.

This would be nice, we agree. The problem is that the model's vertical resolution is only 400 m, and not 10 m like the soundings. To test if our hodograph analysis may work on such coarse data, we thinned out the sounding data in the vertical to match the model resolution, and applied the algorithm. However, the results for wavelengths and periods differed by factors of 2 to 4 from those obtained for the 10 m data. Therefore, we concluded that a resolution of 400 m is insufficient. We added this information to the text at line 181: "The hodograph analysis is not applied to the numerical data, because

we found the vertical resolution to be too coarse."

Technical Corrections:

Line 80: Missing words after "data"

Thank you. Corrected at line 81.

Line 114: should be > 2

Thank you. Corrected at line 115.

Line 235: Fig.6 -> fig.7

Thank you. Corrected at line239.

Reviewer #2

This is an interesting and well written manuscript. The authors use observations collected by a sounding network to diagnose profiles of area-averaged mesoscale divergence and vorticity. Particularly, they show that the observed divergence magnitudes scale approximately inversely with the area equivalent radius, a relationship which is also confirmed in ERA5 reanalysis, and in a numerical simulation. Based on a series of assumptions, it is shown that in the numerical data, the energy spectra of inertia gravity (IG) waves might explain this scaling relationship. This paper is relevant and important for research on tropospheric mesoscale divergence and gravity waves, and their possible relationships and it fits for WCD. I have a few general points and some minor revisions to suggest.

We thank the reviewer for their positive remarks and the comments.

General comments: - The two most interesting findings of the paper are 1) the scaling of divergence amplitude with area equivalent radius, and 2) the possible explanation of this scaling by the wavenumber dependent IG energy spectra. While the former is robust and is confirmed in observations, reanalysis, and numerical simulation, the

latter is based on a series of assumptions and is shown to be held only in the numerical data. I expect the authors to make this point clear in the text and, in particular, justify why this relationship does not hold in observations and reanalysis data.

Please note that there are no direct observations that would allow the computation of 3D modal IG spectra. As for the reanalysis, we clarified our choice to look at ICON at line 261: "We did not perform any calculation for ERA5, as ERA5 clearly lacks energy at small scales and the ICON simulation is more realistic in this regard."

- In the energy spectra of the numerical simulation, fitting a line with a slope of -2 on a very short segment of the spectral line (i.e., k=100-150) is not convincing, and is not justified in the text.

This slight change of slope is now discussed at lines 261-265: "Since it is not clear if the change in spectral slope from k-5/6 to k-2 at k=100 is physical, we repeat the calculation using a slope of k-5/6 for all k>8. This results in the dashed magenta lines of Fig. 6b,d,f. Now, for R<392 km there is no longer a drop in amplitude, but amplitudes remain approximately constant. This behavior better matches the observations (Fig. 5)."

- Among the assumptions that have been made to derive the analytic relationship between energy spectra and divergence amplitudes, I am less convinced that IG waves should mainly propagate zonally. The authors show in section 3.2 that these waves are on average propagating eastward, however, it is also shown that there could be up to 30 deg difference between their results from the ships and HALO (why?).

We thank the reviewer for highlighting this and added a physical explanation at line 249: "Near-zonal propagation is also expected, because IG waves can become equatorially trapped (Wheeler and Kiladis, 1999) and it is these equatorially trapped east- or westward propagating modes that we identify with the normal mode decomposition."

- The authors propose that they will use the global energy spectra to explain why ERA5

has larger divergence amplitudes than ICON but it is not addressed in the paper. ERA5 also shows much larger variability in divergence amplitude at a given equivalent radius (Fig. 6).

We thank the reviewer for noticing the accidentally omitted explanation. We added Figure 8 and the following text at line 239: "Figure 8 shows the energy contained in IG modes in ERA5 relative to ICON. For k<100 the energy in ERA5 exceeds the energy in ICON, and vice versa for k>100. Most of the divergence variability considered here can be attributed to the more energetic large scale, i.e. k<100, explaining the larger divergence amplitudes in ERA5."

Specific comments:

Line 80: data the surface -> data "cover" the surface

Thank you. Corrected at line 81.

Line 114: It should be L/R > 2.

Thank you. Corrected at line 115.

Line 220: thick dots -> thick blue dots

Added at line 222.

Line 237: I think you mean Fig. 7b instead of Fig. 6b.

Thank you. Corrected at line239.

Line 245: Add "of IG modes" after spectral slopes to make it clear.

Added at line 251.

Figure 6: Explain the magenta lines in the caption of the figure. I think it would be great if you could add a secondary x axis to this figure which shows zonal wavenumber. I assume part of the magenta line is based on $\sigma(k)$=-5/6, and part of it based on $\sigma(k)$=-1. Perhaps you could make it clear by using different color/line style. A very good suggestion. The second axis was added, as well as an explanation of the magenta lines. From equation 10, one can see that all larger scales contribute to the smaller scales of divergence. Hence, using different line styles to distinguish what slopes contribute would be misleading. But we believe this is quite clear now with the second axis and the additional analysis (dashed line).

Table 1: Please either add the column number to the table or add the column symbol to the caption.

We added the column symbol to the caption.